# Increasing rice yield with low ammonia volatilization by combined application of controlled-release blended fertilizer and densification

**Xiaowei Ma**[1,2,3], **Zijuan Ding**[1], **Ren Hu**[1], **Xuexia Wang**[2,3], **Jun Hou**[1]*, **Guoyuan Zou**[2,3], **Bing Cao**[2,3]*

**1** MARA Key Laboratory of Sustainable Crop Production in the Middle Reaches of the Yangtze River (Co-construction by Ministry and Province), College of Agriculture, Yangtze University, Jingzhou, China, **2** Institute of Plant Nutrition, Resource and Environment, Beijing Academy of Agricultural and Forestry Sciences, Beijing, China, **3** Beijing Engineering Technology Research Center for Slow/Controlled-Release Fertilizer, Beijing, China

* houjungoodluck1@163.com (JH); caobing@baafs.net.cn (BC)

## Abstract

Controlled-release blended fertilizer (CRBF) and densification can increase rice yield and nitrogen (N) efficiency. However, the effects of CRBF combined with densification on rice yield, N absorption, economic benefits of fertilization, and ammonia volatilization loss remain unclear. A 2-year field experiment was conducted using five treatments: no N (control, CK), conventional N application (farmer's fertilization practice, FFP), optimal N application (OPT), single basal application of CRBF (CRBF), and CRBF combined with densification (CRFDP). Moreover, rice yield, N absorption and use efficiency, economic benefit, and ammonia volatilization loss were evaluated. CRBF and CRFDP significantly increased rice dry matter, N use efficiency by 11.6%–30.5% and 90.2%–160.0%, finally increased the yield by 33.3% and 26.1% in 2021 and 2022, respectively. Compared with FFP, CRFDP with 16.7% reduction of N input significantly increased yield by 33.3% and 26.1% and economic benefit by 46.9% and 38.3% in 2021 and 2022, respectively. Compared with CRBF, CRFDP increased the total yield by 2.7% and 15.2%, economic benefit by 3.5% and 7.6%, and N absorption efficiency by 10% and 8.3% in 2021 and 2022, respectively. Compared with FFP, CRFDP reduced ammonia volatilization intensity by 62.5% and 60.8%, cumulative ammonia volatilization loss by 46.3% and 50.3% and also lowered $NH_4^+$-N of surface water by 69.0%–93.8% and 57.8%–89.7% in 2021 and 2022, respectively. The combination of CRBF and densification could improve the rice yield, economic benefit, and N use efficiency and reduce ammonia volatilization. These results might provide data and theoretical support for the high yield of rice and a new environmentally friendly and resource-efficient model of rice cultivation.

**Data availability statement:** All relevant data are within the paper and its Supporting information files.

**Funding:** Key R&D plan of Hubei Province (2023BBB082; 2022BBA002) Carbon account accounting and carbon reduction and sequestration technology research of Quzhou city of China (2022-31) Beijing Academy of Agriculture and Forestry Sciences Innovation Capacity Building Special (KJCX20230304) Platform Construction of Beijing Academy of Agriculture and Forestry Sciences (PT2023-47): The funding foundations (Beijing Academy of Agriculture and Forestry Sciences Innovation Capacity Building Special, KJCX20230304; Key R&D plan of Hubei Province, 2023BBB082, 2022BBA002; Platform Construction of Beijing Academy of Agriculture and Forestry Sciences PT2023-47) provided research funds for this study. We have revised the funding foundations and marked it with blue font in the Acknowledgments.

**Competing interests:** The authors have declared that no competing interests exist.

## 1. Introduction

China is the world's largest producer and consumer of rice [1]. Rice production in China needs an estimated increase by 20% by 2030 to meet the demands [2]. Over the past 50 years, rice production has increased by nearly threefold without the need for expanding arable land. Approximately 50% of this increase is attributed to the use of chemical fertilizers, with nitrogen (N) fertilizer being the primary contributor [3]. N is one of the essential nutrient elements necessary for plant growth, which is important for plant photosynthesis and has significant effects on rice growth, yield, and quality [4]. Appropriate dosages of N fertilizer can increase the effective panicle number and grain number per panicle of rice to improve the rice yield [5]. For a long time, farmers have relied on common urea to provide the necessary N for crop growth. However, the easy solubility of common urea in water makes it prone to loss, leading to agricultural pollution. Nitrogen fertilizer applied to farmland will significantly increase the surface water ammonium N concentration [6], and it will be lost through ammonia volatilization ways [7]. Excessive N application increases production costs and aggravates N loss to the environment, resulting in water and air pollution. Therefore, effective measures should be taken to reduce N loss and maintain high crop yield.

Controlled-release fertilizer (CRF) promotes rice yield and reduces ammonia emissions [8]. The nutrient demand of rice in the growing season follows an "S"-shaped curve, with low N demand in the seedling stage, high N demand from the transplanting stage to the heading stage, and low N demand from the heading stage to the maturity stage [9]. Studies have shown that plant photosynthetic efficiency, which is closely related to N use efficiency, is affected by leaf enzyme activity and chlorophyll concentration [10,11]. Enhancing leaf enzyme activity and chlorophyll concentration helps the plants to make better use of N and have higher dry matter accumulation [12]. N application during the early growth period of rice could increase the number of tillers, however the excessive number of tillers not only consumes nutrients can deteriorate the aeration of the plant population, intensifying the occurrence of diseases and pests. Conversely, insufficient N fertilizer can lead to slower tillering and insufficient panicles [13]. Controlled-release urea is considered to be effective to improve crop yield and N use efficiency, and the N application rate is reduced by one third that the rice yield could be increased by 3%–5.9% compared with that of common urea [14]. CRF application is also considered an effective measure to reduce N loss in paddy fields [15]. Among the different N loss pathways monitored in paddy fields, ammonia volatilization constitutes the largest portion (68.98%–75.27%), CRF reduces ammonia volatilization (24.69%–29.54%) [16]. The price of CRF is higher than that of common urea, but its economic benefit is finally improved due to a significant reduction. Appropriate CRF with suitable ratios can increase rice yield and further improve economic benefits [17].

Densification promotes rice yield and N efficiency. Reasonable planting density is essential for achieving high rice yield, which is conducive to the formation of a rice population and yield components, and moderate densification can fully exploit population advantages [18]. Densification increased the heading rate, yield, and N absorption by 29.3%, 17.0%, and 32.4%, respectively [19]. Zhu et al. [20]. demonstrated that densification increases planting density by about 50%, correspondingly reduces N input by about 30%, and increases rice yield and NUE by 0.5–7.4% and 14.3–50.6%, respectively. According to the results from Huang et al. [21], with a 20% reduced N application, densification can reduce ammonium N concentration in field water, thereby reducing ammonia volatilization by 50.3%–70.1%. Appropriately increased densification is a good cultivation technology model to achieve high rice yield and high N use efficiency and reduce N loss [22].

Based on the role of CRF and the yield increase caused by densification with reduced nitrogen, we speculated that combined application of controlled-release blended fertilizer and densification could increase rice yield and reduce ammonia volatile emissions, and improve economic benefits and N input also can be reduced for the high N use efficiency. In this study, the effects of controlled-release blended fertilizer (CRBF) combined with denser planting on rice yield, N use efficiency, and ammonia volatilization were studied through a 2-year field experiment so as to provide a theoretical basis and technical support for exploring a new environmentally friendly and high-yield model of rice planting in China.

## 2. Materials and methods

### 2.1. Experimental site

The experimental site was located at the experimental station of Yangtze University (30°23' 46.68"N, 112°29' 7.71" E) in Jingzhou City, Hubei Province. This region belongs to the east monsoon agricultural climate zone, and the middle and lower reaches of the Yangtze River belong to the north subtropical agricultural climate zone. In 2021 and 2022, the average temperature was 16.5 °C and 18.0 °C, and the accumulated temperature with ≥ 10 °C was 5204.3 °C and 6125.3 °C, respectively, and the average annual precipitation was 1095 and 1076 mm, and the average annual sunshine duration was 1718 and 1794 h, respectively. The soil was a lacustrine water-retention type paddy soil. The basic properties of the soil in this plowing layer were as follows: soil pH, 6.27; total N, 1.26 g·kg⁻¹; total phosphorus, 0.51 g·kg⁻¹; total potassium, 9.51 g·kg⁻¹; alkali-hydrolyzed N, 78.61 mg·kg⁻¹; available phosphorus, 20.75 mg·kg⁻¹; available potassium, 95.51 mg·kg⁻¹; organic matter, 22.31 g·kg⁻¹; and ammonium N, 3.79 mg·kg⁻¹. The distribution of temperature and precipitation during the growth period is shown in Fig 1.

### 2.2. Experimental design

Five treatments were set up: (1) control (CK), in which no N fertilizer was used, only phosphate potassium fertilizer was applied; (2) farmer's fertilization practice (FFP), in which 50% N fertilizer was used as the basal fertilizer and 50% topdressing during the tillering stage; (3) optimized N fertilization (OPT), in which 40% N fertilizer was used as the basal fertilizer, 20% topdressing in the tillering stage, and 40% urea N fertilizer; (4) single basal application of CRBF, in which controlled-release fertilizer accounted for 50% of the total N rate; and (5) CRBF combined with densification (CRFDP). The amount and application method of N fertilizer were the same as that of CRBF. Integrated N management was carried

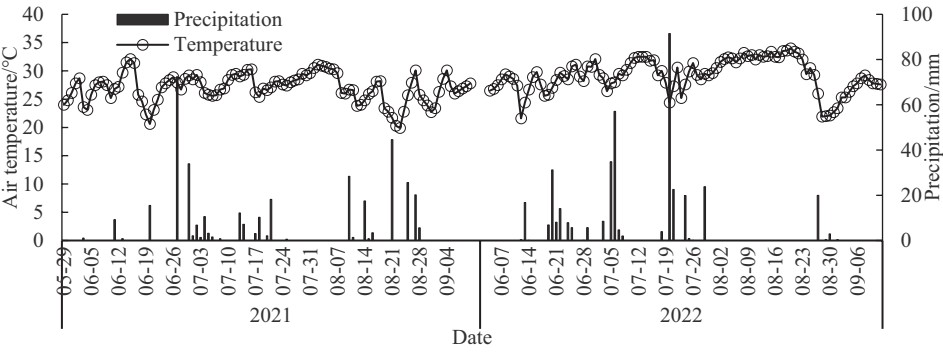

**Fig 1. Precipitation and temperature during rice growing period.**

out with dense planting measures. The row spacing of rice plants in dense cultivation decreased from $20 \times 25\,cm^2$ to $15 \times 25\,cm^2$. The field experiment was arranged following a completely randomized block design with three replicates, 15 experimental plots were used and each plot had an area of $42\,m^2$. N fertilizer was divided into large-particle urea (46%, N) and 4-month coated urea (42% N, produced by the Institute of Plant Nutrition, Resources and Environment, Beijing Academy of Agriculture and Forestry Sciences). Phosphorus ($105\,kg\cdot ha^{-1}$) and potassium ($75\,kg\cdot ha^{-1}$) were the same in each treatment. The phosphate fertilizer used was triple superphosphate ($P_2O_5$, 24%), and the potassium fertilizer used was potassium sulfate ($K_2O$, 50%). Table 1 presents the test scheme. The test site was turned over in winter, and the whole field was tilled about a month before transplanting. After soaking the field, the basal fertilizer was evenly spread and mixed with the soil, and urea was used for topdressing. The rice variety used was Yangxian You 418 (Hefei Fengle Seed Industry Co., Ltd., Hefei, China).

## 2.3. Sample and measurement methods

**2.3.1. Determination of leaf SPAD and N transfer enzyme activities.** At the tillering, booting, and heading stages of rice, 10 functional leaves were randomly selected from each plot, with 5 points taken from each leaf, and the relative chlorophyll content SPAD value was measured using the chlorophyll analyzer SPAD-502 (Konica Minolta, Japan). At these stages, 10 functional leaves were taken from each plot, and the activities of nitrate reductase (NR) and glutamine synthetase (GS) were determined using the in vitro method. The activities of glutamate dehydrogenase (GDH) and glutamate synthetase (GOGAT) were measured using biochemical kits (Suzhou Keming Biotechnology Co., Ltd.).

**2.3.2. Ammonia volatilization and determination of inorganic N content in surface water.** The volatilization of $NH_3$ in the paddy fields was determined using the aeration method [23]. Two sponges coated with glycerol phosphate (2-cm thick and 16-cm diameter) were placed in a polyvinyl chloride (PVC) pipe (16-cm diameter and 25-cm height). The lower sponge absorbed $NH_3$ from the paddy field, and the upper sponge prevented $NH_3$ and dust from entering the air. Three collection devices were set up in each plot. The $NH_3$ samples were collected 1, 3, 5, 7, 9, and 11 days after each fertilization, and the sponge samples were extracted with 300 mL of $1.0\,mol\,L^{-1}$ KCl solution. The Auto Analyzer 3 Continuous-Flow Analysis technique was used to measure the concentration of $NH_4^+$-N (Auto Analyzer 3; Germany).

The rate of $NH_3$ volatilization ($kg\cdot N\cdot ha^{-1}\cdot d^{-1}$) was calculated as follows:

$$NH_3\ \text{flux} = \frac{M}{(A \bullet D) \bullet 10^{-2}}$$

**Table 1. Experimental design.**

| Treatment | Nitrogen application rate/(kg·ha$^{-1}$) | Nitrogen application method | Planting density/($10^4$ ·ha$^{-1}$) |
|---|---|---|---|
| CK | 0 | | 20 |
| FFP | 270 | 50% base fertilizer, 50% tillering fertilizer | 20 |
| OPT | 225 | 40% base fertilizer, 20% tillering fertilizer, 40% spike fertilizer | 20 |
| CRBF | 225 | Application of controlled release blended urea as a single base fertilizer | 20 |
| CRFDP | 225 | Application of controlled release blended urea as a single base fertilizer | 26 |

CK, control, no nitrogen; FFP, Farmer's fertilization practice; OPT, optimized nitrogen application; CRBF, controlled release blended fertilizer; CRFDP, controlled release blended fertilizer combined with dense planting.

where $M$ is the amount of $NH_3$ collected by the sponge with glycerol phosphate, mg; $A$ is the cross-sectional area of the PVC pipe, m²; and $D$ is the interval of $NH_3$ collection, day. The sum of $NH_3$ emission fluxes during the sampling days was used to calculate the cumulative $NH_3$ emission.

$$NH_3 - N\ loss\ (\%) = \frac{N_{cumulative} - N_{background}}{N_{applied-N}} \times 100 \tag{1}$$

where $N_{cumulative}$ is the cumulative ammonia volatilization loss under N application treatment, kg·ha⁻¹; $N_{background}$ is the cumulative ammonia volatilization loss of the control without N application, kg·ha⁻¹; and $N_{appplied}-N$ is the N application rate of the N application treatment, kg·ha⁻¹.

Field water was collected in the tillering, booting, and full heading stages, and the concentrations of ammonium N and nitrate N in the field water were measured using the flow injection analysis (Auto Analyzer 3, German).

**2.3.3. Calculation of yield and N use efficiency.** Three points were selected in each plot for rice harvesting. The common sample frame method was used to avoid the influence of marginal utility. The yield area was 4 m², and the weight was measured by harvesting, threshing, and drying. Five holes of rice plants were selected from each plot, and indoor seed testing was conducted after drying. The aboveground plants were divided into stems, leaves, ears, and grains, placed in an oven for half an hour at 105 °C, dried at 80 °C until the quality remained unchanged, and then weighed. Subsequently, the total N content of the sample was determined using the Kjeldahl method.

The N recovery efficiency (NRE, %) was calculated as follows:

$$NRE = \frac{N_a - N_b}{N_r} \times 100 \tag{2}$$

where $N_a$ represents the aboveground N uptake of plants in the N application area, kg·ha⁻¹; $N_b$ is the aboveground N uptake of plants in the no N application area, kg·ha⁻¹; and $N_r$ is the N application rate in the N application area, kg·ha⁻¹.

The N agronomic efficiency (NAE, kg·kg⁻¹) was calculated as follows:

$$NAE = \frac{Y_a - Y_b}{N_r} \tag{3}$$

where $Y_a$ represents the yield of the N application area, kg·ha⁻¹; and $Y_b$ is the yield in the no N application area, kg·ha⁻¹.

The N partial factor productivity (NPFP, kg·kg⁻¹) was calculated as follows:

$$NPFP = \frac{Y}{N_r} \tag{4}$$

where $Y$ represents the total yield, kg·ha⁻¹.

The N physiological efficiency (NPE, kg·kg⁻¹) was calculated as follows:

$$NPE = \frac{Y_a - Y_b}{N_a - N_b} \tag{5}$$

where $Y_a$ represents the grain yield under N application treatment, kg·ha⁻¹; and $Y_b$ is the grain yield no N application treatment, kg·ha⁻¹.

The dry matter exportation from vegetative organs (DME, t·ha$^{-1}$) was calculated as follows:

$$DME = V_a - V_b \qquad (6)$$

where $V_a$ is the dry weight of nutrient organs during the full heading period, t·ha$^{-1}$; and $V_b$ is the dry weight of mature vegetative organs, t·ha$^{-1}$.

The transportation rate of dry matter from vegetative organs (TRDV, %) was calculated as follows:

$$TRDV = \frac{V_a - V_b}{V_a} \times 100 \qquad (7)$$

**2.3.4. Economic benefit accounting.** The cost profit analysis was based on the method described by Gou et al. (2017).

The net income (NI, yuan·ha$^{-1}$) was calculated as follows:

$$NI = RY_F \times P_r - C_t \qquad (8)$$

where $RY_F$ is the rice yield, kg·ha$^{-1}$; $P_r$ is the unit price of rice, yuan·kg$^{-1}$; and $C_t$ is the total cost, yuan·ha$^{-1}$.

The total cost ($C_t$, yuan·ha$^{-1}$) was calculated as follows:

$$C_t = C_a + C_l + C_s \qquad (9)$$

where $C_a$ is the cost of agricultural materials, yuan·ha$^{-1}$; $C_l$ is the labor cost, yuan·ha$^{-1}$; and $C_s$ is the cost of seedlings, yuan·ha$^{-1}$.

The production investment ratio (PIR) was calculated as follows:

$$PIR = \frac{IN_t}{C_t} \qquad (10)$$

where $IN_t$ is the total income, yuan·ha$^{-1}$.

## 2.4. Data process

SPSS 19.0 (IBM, Inc., Armonk, NY, USA) was used for a one-way analysis of variance (ANOVA), and significant differences ($p < 0.05$) between the treatments are indicated by different letters. All the data diagrams were prepared using Microsoft Excel 2010 (Redmond, WA, USA), and the correlation between NH$_3$ volatilization and the concentration of NH$_4^+$-N in field water and the correlation between NH$_3$ volatilization and concentration of NH$_4^+$-N in soil were analyzed. Diagrams for the diffusion of NH$_4^+$-N in the soil were prepared using Surfer 8.0 (Surface mapping system; Golden Software, Inc., Golden, CO, USA).

## 3. Results

### 3.1. SPAD, N transferases, and dry matter transport

N application significantly increased the relative chlorophyll content of rice leaves in different growth stages ($p < 0.05$) (Table 2). The SPAD value was similar for CRBF and CRFDP treatments and FFP and OPT treatments in the tillering and full heading stages, whereas the SPAD value of the leaves exhibited the following trend CRBF = CRFDP> FFP> OPT in the booting stage. The CRBF significantly increased the relative chlorophyll content of rice leaves.

Fertilization significantly increased the activities of GDH, glutamate synthase (GOGAT), glutamine synthase (GS), and NR in rice leaves (Fig 2). The activity of GDH in rice leaves

**Table 2.** SPAD of rice leaves under different treatments.

| Year | Treatment | Tillering stage | Booting stage | Full heading stage |
|------|-----------|-----------------|---------------|--------------------|
| 2021 | CK | 38.44 ± 0.56c | 36.34 ± 0.47c | 38.51 ± 0.92b |
|      | FFP | 44.64 ± 0.54a | 39.00 ± 0.64b | 43.23 ± 1.19a |
|      | OPT | 41.73 ± 0.33b | 37.71 ± 0.14b | 43.34 ± 1.96a |
|      | CRBF | 43.26 ± 1.56ab | 41.20 ± 1.15a | 43.13 ± 2.51a |
|      | CEFDP | 43.58 ± 1.41ab | 41.47 ± 0.75a | 41.91 ± 0.50a |
| 2022 | CK | 34.73 ± 1.95c | 31.50 ± 1.05c | 34.80 ± 0.78b |
|      | FFP | 41.30 ± 0.30a | 38.23 ± 0.35b | 40.77 ± 0.42a |
|      | OPT | 37.90 ± 0.80b | 38.03 ± 0.74b | 41.53 ± 0.58a |
|      | CRBF | 41.70 ± 0.53a | 42.17 ± 0.38a | 42.47 ± 0.81a |
|      | CEFDP | 41.50 ± 0.95a | 40.87 ± 1.24a | 42.20 ± 0.46a |

Values followed by different small letters within each column are significantly different among treatments ($p < 0.05$).

gradually increased with the growth period and reached its highest point in the full heading stage. At the tillering stage, FFP had the higher GDH activity than the other three N application treatments. At the booting and full heading stages, CRBF had significantly higher GDH activity than the other three N application treatments. The GOGAT activity of rice leaves gradually increased with the growth period and reached its highest point in the full heading stage. At the tillering stage, no significant difference was found in the GOGAT activity among the different N treatments. However, the GOGAT activity of CRBF was higher than that of the other three N application treatments at the booting and full heading stages. The GS activity of rice leaves gradually increased with the growth period and reached its highest point at the full heading stage. The GS activity of FFP was higher than that of the other three N application treatments in the tillering and booting stages, while GS of CRBF was higher than that of the other three N application treatments at the full heading stage. The NR activity was positively correlated with nitrate concentration in plants. At the tillering and booting stages, FFP had higher NR activity than the other three optimized N applications. At the full heading stage, the CRBF and CRFDP was higher NR activity than FFP and OPT. No significant difference was found between the two CRBF treatments.

Accumulation of dry matter in rice and the transport of dry matter in nutrient organs were improved by N fertilization significantly (Table 3). At the tillering, booting, full heading, and mature stages, the N treatments had significantly higher dry matter accumulation than CK. Compared with FFP, CRBF reduced dry matter accumulation by 6.4% and 2.9% at the tillering and booting stages, respectively, and increased dry matter accumulation by 0.9% and 11.2% at the full heading and mature stages, respectively. Compared with OPT, CRBF increased dry matter accumulation by 3.4%, 12.9%, 6.3%, and 11.6% at the tillering, booting, full heading, and mature stages, respectively. Compared with CRBF, CRFDP increased dry matter accumulation by 12.2%, 7.8%, 13.1%, and 16.9% at the tillering, booting, full heading, and mature stages, respectively. Compared with other N treatments, the dry matter output of nutrient organs was higher and the harvesting index was lower under CRFDP. Taken together, CRBF and CRFDP increased the accumulation of dry matter in rice and the output of dry matter from nutrient organs during the filling period to achieve high rice yield.

## 3.2. Rice yield and N fertilizer use efficiency

Significant differences were found in the yield components of rice among different treatments, and N application significantly increased the number of effective panicles and filled grains per

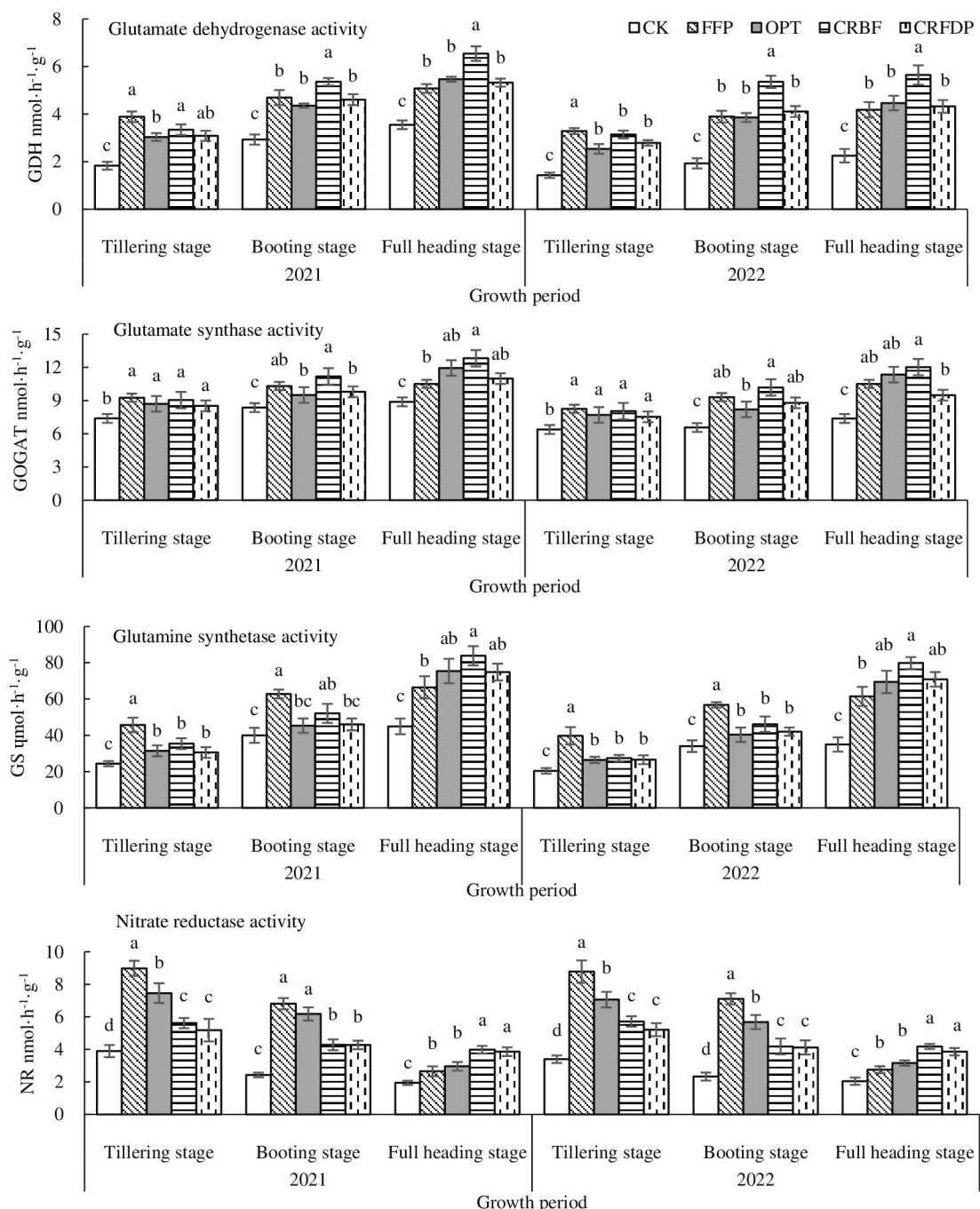

**Fig 2. Rice leaf enzyme activity under different treatments.**

panicle ($p < 0.05$). The thousand-grain quality of rice was the lowest after FFP, whereas other treatments showed no significant difference (Table 4). Compared with CK, N fertilization treatments significantly increased the effective panicle number by 88%–149% and 44%–84% ($p < 0.05$) and the number of grains per panicle by 31.6%–56.2% and 33.1%–51.6% in 2021 and 2022 ($p < 0.05$), respectively. Compared with FFP, CRFDP significantly increased the

**Table 3. Dry matter accumulation, transport and harvest index of rice under different treatments in 2022.**

| Treatment | Dry matter accumulation (t·ha⁻¹) | | | | RDMA-FHM (%) | DME (t·ha⁻¹) | TRDV (%) | Harvest index (%) |
|---|---|---|---|---|---|---|---|---|
| | Tillering stage | Booting stage | Full heading stage | Maturation stage | | | | |
| CK | 1.17c | 4.48c | 7.84d | 13.12c | 39.85a | 1.00d | 24.90a | 51.18ab |
| FFP | 2.62ab | 8.55a | 14.02b | 19.77b | 29.07d | 2.11bc | 22.05ab | 51.01ab |
| OPT | 2.37b | 7.35b | 13.30c | 19.68b | 32.42c | 1.85c | 19.83b | 53.45a |
| CRBF | 2.45ab | 8.30ab | 14.15b | 21.97ab | 35.61b | 2.28ab | 21.08b | 50.29b |
| CRFDP | 2.75a | 8.95a | 16.30a | 25.68a | 36.54b | 2.53a | 22.73ab | 49.52b |

Values followed by different small letters within each column are significantly different among treatments ($p < 0.05$).

**Table 4. Rice yield and yield components under different treatments.**

| Year | Treatment | Effective panicle/(10⁴·ha⁻¹) | grains per panicle (No.) | 1000-grain mass (g) | Yield (kg·ha⁻¹) |
|---|---|---|---|---|---|
| 2021 | CK | 142.22 ± 5.90c | 158.07 ± 1.30d | 25.37 ± 0.86a | 8106.00 ± 30.28d |
| | FFP | 280.00 ± 6.71b | 208.07 ± 5.08c | 23.06 ± 0.40b | 10028.93 ± 213.68c |
| | OPT | 268.89 ± 11.08b | 216.87 ± 8.99c | 24.74 ± 0.76a | 11101.47 ± 218.44b |
| | CRBF | 266.67 ± 6.73b | 232.13 ± 3.16b | 25.44 ± 0.25a | 13015.00 ± 152.35a |
| | CRFDP | 353.77 ± 5.88a | 246.93 ± 8.48a | 25.48 ± 0.59a | 13370.48 ± 203.33a |
| 2022 | CK | 192.00 ± 16.00c | 160.07 ± 3.16d | 24.02 ± 0.63a | 6713.93 ± 349.45d |
| | FFP | 277.33 ± 9.24b | 213.93 ± 3.78c | 22.84 ± 0.99b | 10085.33 ± 485.34c |
| | OPT | 288.00 ± 16.00b | 225.27 ± 4.15b | 24.19 ± 0.45a | 10519.33 ± 534.65bc |
| | CRBF | 293.33 ± 9.24b | 230.33 ± 4.12b | 24.45 ± 0.38a | 11046.50 ± 383.92b |
| | CRFDP | 352.67 ± 13.28a | 242.67 ± 5.99a | 24.60 ± 0.52a | 12720.83 ± 423.09a |
| Variance analysis | | | | | |
| Year (Y) | | ** | ** | ** | ** |
| Fertilization (F) | | ** | ** | * | ** |
| Y × F | | ** | NS | NS | ** |

Values followed by different small letters within each column are significantly different among treatments ($p < 0.05$). NS, no significant difference; * $p < 0.05$; ** $p < 0.01$.

number of effective panicles by 26.1% and 27.2% and grains per panicle by 18.7% and 13.4%, in 2021 and 2022 ($p < 0.05$), respectively. N application significantly increased rice yield by 23.7%–64.9% and 50.2%–89.4% in 2021 and 2022 ($p < 0.05$), respectively, while CRBF and CRFDP significantly increased rice yield by 17.3%–33.3% and 5.1%–26.1% in 2021 and 2022, respectively, compared with FFP and OPT ($p < 0.05$). Compared with CRBF, CRFDP increased rice yield by 2.7% ($p > 0.05$) and 15.2% ($p < 0.05$) in 2021 and 2022, respectively. The highest rice yield with low N input was noted under CRFDP and CRBF treatments in 2021 and 2022, respectively, indicating that the combined application of CRBF and densification could improve rice yield. The decrease in yield in 2022 was mainly due to the prolonged period of non-precipitation during the rice flowering period, when pollination of rice flowers was impeded.

Fertilization methods can significantly affect N use efficiency in rice (Table 5). Compared with FFP, OPT, CRBF and CRFDP had higher N use efficiency. In 2021 and 2022 yr, N use efficiency was increased by 55.8%–160.0% and 24.6%–90.2% ($p < 0.05$), NPFP was significantly increased by 12.2–22.3 and 9.4–19.2 kg·kg⁻¹ ($p < 0.05$), NAE was significantly increased by 6.2–16.3 and 4.4–14.2 kg·kg⁻¹ ($p < 0.05$), and NPE was significantly increased by 5.5–13.0 and 4.8–10.4 kg·kg⁻¹ ($p < 0.05$). Compared with CRBF, CRFDP showed higher N use

**Table 5. Nitrogen use efficiency under different treatments.**

| Year | Treatment | N use efficiency/% | N partial factor productivity/kg·kg⁻¹) | N agronomic efficiency/kg·kg⁻¹) | N physiological efficiency/(kg·kg⁻¹) |
|------|-----------|-------------------|------------------------------------------|----------------------------------|--------------------------------------|
| 2021 | FFP | 21.5 ± 2.4d | 37.14 ± 0.79c | 7.12 ± 0.73c | 34.34 ± 6.14b |
| | OPT | 33.5 ± 2.6c | 49.34 ± 0.97b | 13.31 ± 0.91b | 39.87 ± 2.00ab |
| | CRBF | 46.1 ± 1.2b | 57.84 ± 0.68a | 21.82 ± 0.66a | 47.36 ± 0.18a |
| | CRFDP | 56.1 ± 3.9a | 59.42 ± 0.90a | 23.39 ± 0.77a | 42.04 ± 2.71ab |
| 2022 | FFP | 28.5 ± 2.8d | 37.35 ± 1.79c | 12.48 ± 0.73c | 36.16 ± 2.83c |
| | OPT | 35.5 ± 2.3c | 46.75 ± 2.37b | 16.91 ± 0.91b | 40.93 ± 3.17b |
| | CRBF | 45.9 ± 2.0b | 49.09 ± 1.71a | 25.25 ± 0.66a | 42.79 ± 1.86ab |
| | CRFDP | 54.2 ± 1.5a | 56.53 ± 1.88a | 26.69 ± 0.77a | 46.56 ± 2.45a |

Values followed by different small letters within each column are significantly different among treatments ($p < 0.05$).

efficiency, which was significantly increased by 21.7% and 18.1% ($p < 0.05$) in 2021 and 2022, respectively. No significant difference was found in NPFP, NAE, and NPE.

### 3.3. Ammonia volatilization in rice fields and inorganic N content in surface water

After applying basal fertilizers, N application treatment showed a peak of ammonia volatilization on the first day after fertilization. The ammonia volatilization rate was significantly higher after the N treatment than after the CK ($p < 0.05$) (Fig 3), and the highest value was obtained after the FFP. Ammonia volatilization was continued for 13 days in this stage. After applying tillering fertilizers, a peak in ammonia volatilization was also experienced after all N application treatments on the first day. At this stage, ammonia volatilization continued for 11 days, while it was the highest under FFP, and the value of CRBF and CRFDP was significantly decreased to lower levels on the fifth day after fertilization. After applying ear fertilizers, significant ammonia volatilization loss was observed under OPT, while the ammonia volatilization flux of other treatments was very low, which lasted for 9 days.

Compared with CK treatment, N fertilizer significantly increased the ammonium N concentration in paddy surface water in the growth period, however, CRBF and CRFDP were particular cases in the tillering stage in 2021 (Fig 4). At the tillering stage, the ammonium N concentration of FFP was significantly higher in the surface water than those of OPT, CRBF, and CRFDP ($p < 0.05$). At the booting and full heading stages, FFP and OPT had the significantly higher ammonium N concentration than CRBF and CRFDP ($p < 0.05$).

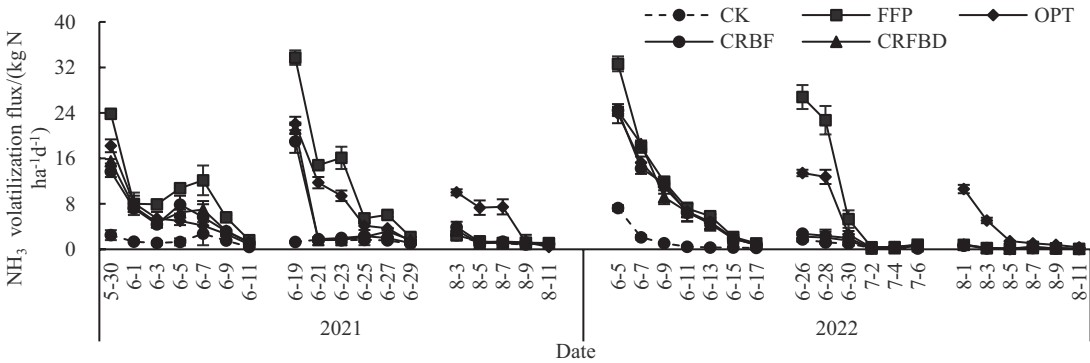

**Fig 3. Dynamics of NH₃ volatilization fluxes during rice growing period under different treatments.**

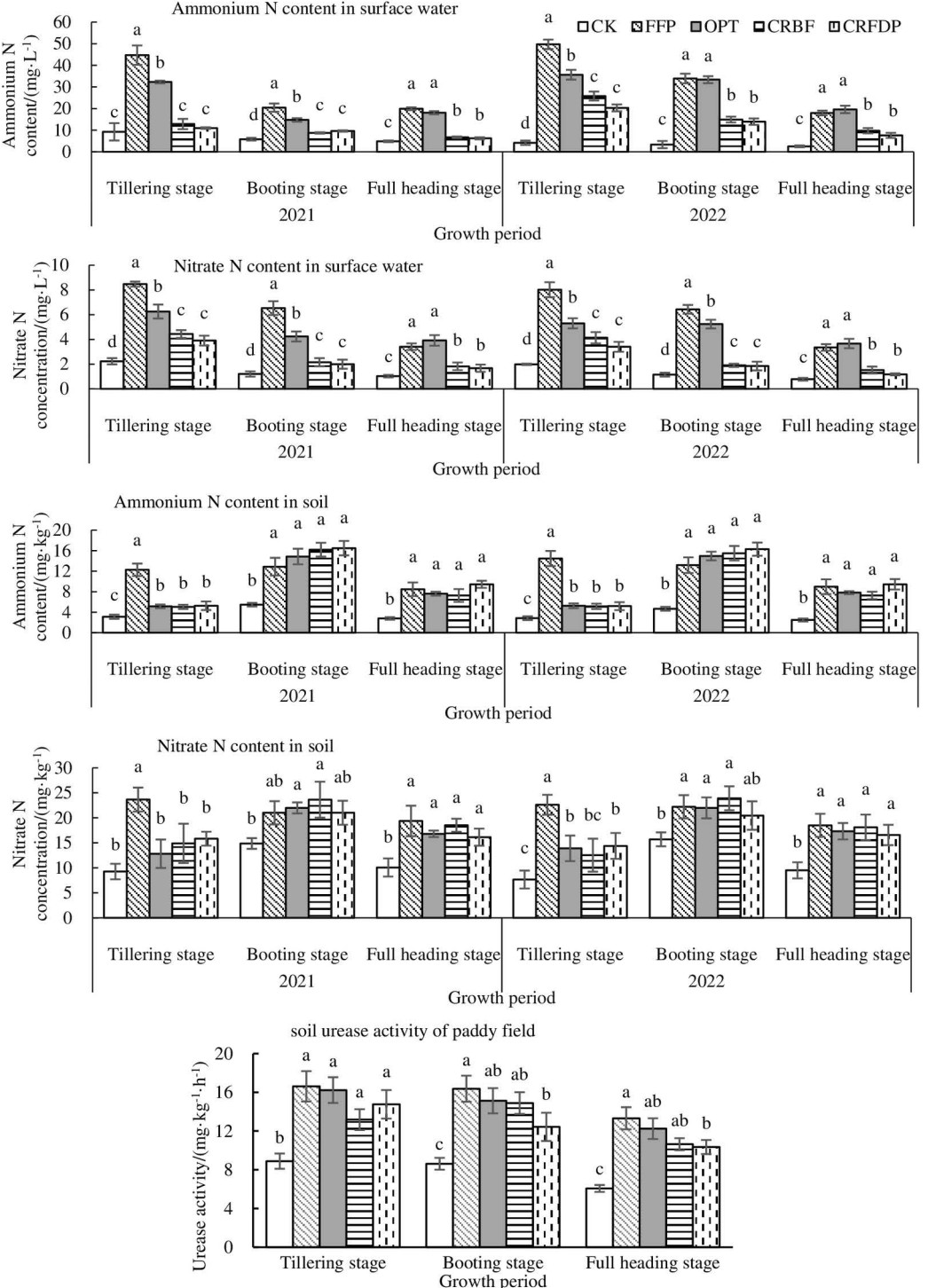

**Fig 4. Ammonium N and nitrate N content in surface water and soil and soil urease activity of paddy field under different treatments.**

Compared with CK, N application significantly increased the nitrate N concentration in the field water during the rice growth period ($p < 0.05$) (Fig 4). At the tillering and booting stages, FFP had significantly higher nitrate N concentration in the surface water than OPT, CRBF, and CRFDP ($p < 0.05$). During the entire growth period of rice, FFP and OPT had the significantly higher concentration of nitrate N was in the surface than CRBF and CRFDP ($p < 0.05$).

Compared with CK, N application significantly increased soil ammonium N concentration ($p < 0.05$) (Fig 4). At the tillering stage, soil ammonium N concentration of FFP was significantly higher than those of OPT, CRBF and CRFDP ($p < 0.05$). N application significantly increased soil nitrate N concentration ($p < 0.05$) (Fig 4). At the tillering stage, soil nitrate N concentration of FFP was significantly higher than those of OPT, CRBF and CRFDP ($p < 0.05$).

Compared with CK, N application significantly increased soil urease activity ($p < 0.05$) (Fig 4). The urease activity of CRFDP was significantly lower than that of FFP at the booting and full heading stages ($p < 0.05$). This was consistent with the change trend of ammonium N and nitrate N contents in surface water.

The correlation analysis showed that $NH_4^+$-N content in surface water and soil was positively correlated with the rate of $NH_3$ volatilization ($R^2 > 0.5$) (Fig 5). After urea application, the peak of $NH_3$ emission appeared quickly, mainly due to the rapid hydrolysis of urea into $NH_4^+$-N under the action of urease [24], and the accumulation of $NH_4^+$-N led to the volatilization of $NH_3$. This suggested that the low $NH_3$ emissions from CRBF were mainly due to the lower $NH_4^+$-N content in surface water and soil.

In the basal fertilizer stage, N application treatments had the significantly higher $NH_3$ volatilization loss than CK ($p < 0.05$), and FFP had the significantly higher ammonia volatilization loss than the other three fertilizer treatments. At the tillering stage, the $NH_3$ volatilization loss of FFP was still significantly higher than those of the other three fertilizer treatments, while the $NH_3$ volatilization loss was significantly higher after OPT treatment than those after CRBF and CRFDP treatments. At the panicle fertilization stage, the $NH_3$ volatilization loss of OPT was the highest, followed by FFP and OPT, and the lowest were those of CRBF and CRFDP. The $NH_3$ volatilization rate and $NH_3$ volatilization loss at different fertilization stages were consistent (Fig 3 and Table 6), indicating that N application methods had significantly impact on $NH_3$ volatilization loss.

N application significantly increased the $NH_3$ volatilization intensity ($p < 0.05$). Compared with FFP, OPT, CRBF and CRFDP decreased the $NH_3$ volatilization by 26.87%, 31.6% and

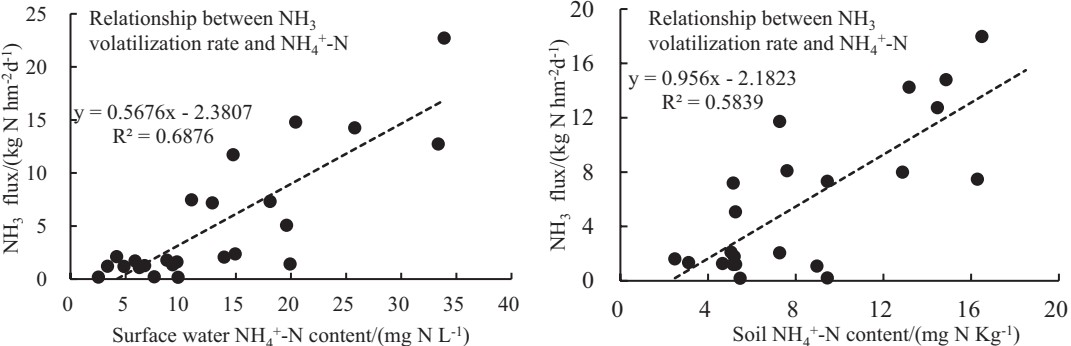

**Fig 5. Relationship between $NH_3$ volatilization rate and $NH_4^+$-N content in surface water and soil.**

60.73% in 2021 yr, and 53.1%, 62.5% and 60.8% in 2022 yr (Table 7). Compared with CK, the cumulative $NH_3$ volatilization loss of N treatments was significantly increased by 51.17–122.64 kg·ha⁻¹ and 42.19–110.02 kg·ha⁻¹ in 2021 and 2022, respectively, after N treatment, and it was the highest after FFP treatment. Compared with FFP, OPT, CRBF, and CRFDP significantly reduced $NH_3$ volatilization loss by 19.0%–17.3%, 49.0%–43.7%, and 46.3%–50.3% in 2021–2022, respectively. The emission reduction effect was better after CRBF and CRFDP treatments than after OPT treatment at the same N rate with 225 kg N ha⁻¹. The $NH_3$ volatilization loss rate was as high as 45.4%–40.7% after FFP in 2021–2022. The $NH_3$ volatilization loss rate of OPT was lower than that of FFP. OPT had similar $NH_3$ volatilization loss rate than FFP. CRBF and CRFDP significantly reduced the ammonia volatilization loss rate to 22.7%–24.4% and 18.8%–22.4% than FFP, respectively. Therefore, optimized N application, especially CRBF could significantly reduce the ammonia volatilization intensity and the environmental harm caused by $NH_3$ volatilization loss. It can be seen from the variance analysis that the year and

**Table 6. Ammonia volatilization loss under different treatments.**

| Year | Treatment | Base fertilizer/ (kg·ha⁻¹) | Tillering fertilizer/ (kg·ha⁻¹) | Panicle fertilizer/ (kg·ha⁻¹) |
|---|---|---|---|---|
| 2021 | CK | 9.01 ± 1.95c | 7.80 ± 0.99d | 6.52 ± 0.44c |
| | FFP | 62.63 ± 4.85a | 76.10 ± 1.43a | 7.25 ± 0.44bc |
| | OPT | 40.90 ± 3.16b | 51.06 ± 0.65b | 26.46 ± 1.23a |
| | CRBF | 38.79 ± 2.39b | 27.04 ± 1.27c | 8.67 ± 1.91b |
| | CRFDP | 41.15 ± 0.67b | 29.79 ± 0.83c | 7.45 ± 1.01bc |
| 2022 | CK | 12.89 ± 0.58b | 4.63 ± 0.44c | 1.38 ± 0.39c |
| | FFP | 64.70 ± 2.31a | 56.41 ± 0.95a | 7.82 ± 0.27b |
| | OPT | 53.25 ± 4.08a | 28.18 ± 0.96b | 20.28 ± 1.73a |
| | CRBF | 59.29 ± 3.14a | 8.36 ± 1.23c | 1.59 ± 0.38c |
| | CRFDP | 53.68 ± 4.58a | 5.71 ± 0.54c | 1.72 ± 0.34c |

Values followed by different small letters within each column are significantly different among treatments ($P < 0.05$).

**Table 7. Ammonia volatilization loss of rice growing period in paddy field.**

| Year | Treatment | $NH_3$ volatilization intensity/(kg·t⁻¹) | Cumulation volatilization emission/(kg·ha⁻¹) | Ammonia volatilization loss rate/% |
|---|---|---|---|---|
| 2021 | CK | 2.88 ± 0.17d | 23.34 ± 2.29d | – |
| | FFP | 14.59 ± 0.80a | 145.98 ± 8.53a | 45.4a |
| | OPT | 10.67 ± 0.22b | 118.42 ± 2.05b | 42.3a |
| | CRBF | 5.73 ± 0.12c | 74.51 ± 2.68c | 22.7b |
| | CRFDP | 5.47 ± 0.15c | 78.39 ± 3.03c | 24.4b |
| 2022 | CK | 2.82 ± 0.12d | 18.91 ± 0.59d | – |
| | FFP | 12.78 ± 0.22a | 128.93 ± 10.93a | 40.7a |
| | OPT | 9.67 ± 0.44b | 101.71 ± 9.55b | 36.8b |
| | CRBF | 6.27 ± 0.42c | 69.24 ± 7.87c | 22.4c |
| | CRFDP | 4.80 ± 0.30c | 61.10 ± 4.68c | 18.8d |
| Variance analysis | | | | |
| Year(Y) | | * | ** | ** |
| Fertilization(F) | | ** | ** | ** |
| Y × F | | NS | ** | ** |

Values followed by different small letters within each column are significantly different among treatments ($p < 0.05$); NS, no significant difference; * $p < 0.05$; ** $p < 0.01$.

treatment had significant effects on the accumulation of NH$_3$ volatilization. The main reason for the significant effect of the year was that there was no precipitation for a long time from July 26 to August 25, 2022 (Fig 1).

### 3.4. Economic benefit evaluation of rice

Compared with CK, N fertilization increased yield output value per hectare by 3198–12611 and 6380–13815 Yuan (Table 8), with an increase of 18.9%–74.7% and 48.9%–105.8% in 2021 and 2022, respectively. Compared with FFP, OPT, CRBF, and CRFDP increased the income by 2888–2292, 8421–5541, and 9413–7435 Yuan in 2021–2022, respectively, with 16.7% N reduction. CRFDP had the highest output value and NI. Compared with CRBF, CRFDP treatment increased the income per hectare by 992 and 1896 Yuan in 2021 and 2022, respectively. In conclusion, the combination of CRBF and densification maximized the yield and economic benefit.

The cost of agricultural materials, labor, and seedlings is shown in S1 Table from Supporting Information.

## 4. Discussion

### 4.1. Effects of CRFDP on rice growth

CRF continuously supplies N to meet the N demand during the growth period of rice [25]. At the same time, densification increases the rice root system and can more fully absorb inorganic N, increase the number of rice tillers 18.16–19.24%, increase SPAD 2.54–3.38%, ultimately increase rice yield 6.01–7.71% [26,27]. N plays an important role in regulating crop chlorophyll synthesis, and N supply level is significantly positively correlated with the crop chlorophyll content [28,29]. This study showed that SPAD values (Table 2) were not increased by CRBF and CRFDP at the tillering and full heading stages, while the values were increased by 5.6%–10.9% at the booting stage ($p < 0.05$). The SPAD of CRFDP was lower than that of CRBF, indicating that densification reduced the chlorophyll content of rice leaves. This was because that the chlorophyll content of them was decreased with crops lacked N [30]. In the late growth period of rice (e.g., booting stage), the chlorophyll of rice leaves was gradually decreased due to leaf aging and leaf fibrosis [31]. For different N levels, the chlorophyll content was increased with the increase of the N application rate, it reached a

**Table 8. Economic benefits of different treatments.**

| Year | Treatment | Output value/ (yuan·ha⁻¹) | Fertilizer cost/ (yuan·ha⁻¹) | Pesticide costs/ (yuan·ha⁻¹) | Seedling cost/ (yuan·ha⁻¹) | Labor input/ (yuan·ha⁻¹) | Net income/ (yuan·ha⁻¹) | Output/ input |
|---|---|---|---|---|---|---|---|---|
| 2021 | CK | 22696 d | 799 | 700 | 1350 | 2960 | 16887 d | 3.91 b |
| | FFP | 28081 c | 2266 | 700 | 1350 | 3680 | 20085 c | 3.51 c |
| | OPT | 31084 b | 2021 | 700 | 1350 | 4040 | 22973 b | 3.83 b |
| | CRBF | 36442 a | 2566 | 700 | 1350 | 3320 | 28506 a | 4.59 a |
| | CRFDP | 37434 a | 2566 | 700 | 1755 | 3320 | 29498 a | 4.72 a |
| 2022 | CK | 18866 d | 799 | 700 | 1350 | 2960 | 13057 e | 3.25 c |
| | FFP | 27403 c | 2266 | 700 | 1350 | 3680 | 19437 d | 3.44 bc |
| | OPT | 29840 b | 2021 | 700 | 1350 | 4040 | 21729 c | 3.68 b |
| | CRBF | 32914 a | 2566 | 700 | 1350 | 3320 | 24978 b | 4.15 a |
| | CRFDP | 34808 a | 2566 | 700 | 1755 | 3320 | 26872 a | 4.39 a |

Values followed by different small letters within each column are significantly different among treatments ($p < 0.05$).

significant level in the wax ripening stage and an extremely significant level in other growth stages [32]. Xu et al. [33] found that N supply positively influenced both chlorophyll and N contents, which increased significantly with the increase in the N application rate, indicating that increased N application was conducive to the improvement in N and chlorophyll contents in rice leaves ($p < 0.05$). Zhang et al. [34] showed that at the middle tillering, booting, and heading stages, the chlorophyll content of rice leaves of eight seedlings per hole decreased by 19.37%, 16.01%, 22.47%, and 31.77%, respectively, compared with two seedlings per hole, indicating that the planting density had a significant effect on the chlorophyll content of rice leaves.

NR, GS, and GOGAT were key leaf N assimilation enzymes in plant N metabolism [35]. The results showed that CRF had the significantly lower activities of NR, GS, and GOGAT of functional leaves treated than conventional fertilizer at the tiller and booting stages, but had significantly higher NR, GS, and GOGAT of functional leaves at the full heading stage (Fig 2) ($p < 0.05$). In the whole rice growth stage, NR activities were the highest in the tillering stage and decreased gradually, while GS and GOGAT activities increased gradually and reached the highest in the full heading stage. The activities of NR, GS, and GOGAT in leaves of CRFDP were lower than those of CRBF, indicating that densification could reduce the activities of N metabolizing enzymes in rice. This was because NR activity first affected the N absorption rate and then the yield and quality of crops. Its activity was significantly positively correlated with nitrate concentration in plants ($p < 0.05$). When $NO_3^-$ entered rice, it was further converted into amino acids only after being reduced by NR [36,37]. GS catalyzed $NH_4^+$ and glutamic acid to synthesize glutamine, and GOGAT catalyzed glutamine and α-ketoglutaric acid to form glutamate. GS and GOGAT together constituted the GS/GOGAT cycle [38] that participated in the primary absorption of ammonia and reabsorption of ammonia released by photorespiration, as well as assimilation and N fixation of ammonia. Du et al. [39] showed that CRBF (60% CRU + 40% U) treatment could significantly increase the NR activity in rice leaves by 28.2% and GS activity of rice leaves by 66.7% in the full heading stage compared with common urea from FFP. Xu et al. [40] showed that the activities of GS and GOGAT in leaves were the highest in the heading stage, while NR reached its peak at the end of the tillering stage. Moreover, the activities of N metabolizing enzymes first increased and then decreased with the increase in the N application rate, indicating that high N reduced the activities of N metabolizing enzymes in leaves.

## 4.2. Effects of CRFDP on rice yield and N uptake

The experimental results verified the hypothesis that dense planting combined with controlled-release fertilizer (CRF) can increase yield and N uptake. Single basal application of controlled-release fertilizers and densification are two important cultivation techniques for high-yield rice production. CRF can achieve the dynamic balance of N nutrients in rice by regulating nutrient release patterns [41]. Controlled-release urea continuously releases N in the middle and late stages of rice growth, enhances the role of N fertilizer, and increases N uptake by plants, thus effectively improving N use efficiency [42,43]. In this study, CRBF and CRFDP significantly increased rice yield by 9.5%–26.1% and 29.8%–33.3% compared with FFP, and by 17.3%–20.5% and 5.1%–20.9% compared with OPT in 2021 and 2022, respectively ($p < 0.05$). CRFDP significantly increased NUE by 18.1%–21.7% compared with CRBF ($p < 0.05$). The main reason was that CRFDP significantly increased the effective panicles by 26.1% and 27.2%, and the grains per panicle by 18.7% and 13.4% (2021 and 2022) ($p < 0.05$) compared with FFP. This is because the N supply of CRBF was superior to that of soluble N fertilizers [21]. The replacement of urea by controlled-release urea could significantly increase

the effective panicles and the grains per panicle, which was consistent with the conclusion of related studies that CRBF application could improve the rice yield [44]. Single basal application of controlled-release fertilizers not only reduced the labor input but also lowered the fertilizer consumption by 10%, increased yield by 4.6%, increased N use efficiency by 3.6%, and reduced N loss by 47.6% [45]. Ma et al. [46] showed that rice yield, plant N uptake, N use efficiency, and economic benefit increased by 13.35%, 13.98%, 8.98%, and 3040 Yuan·ha$^{-1}$, respectively, under one-time fertilization with CRBF. Hou et al. [47] showed that controlled-release urea could increase the leaf area index in the middle and late stages of rice by 14.45% and dry matter accumulation by 4.47%($p < 0.05$). While ensuring the number of effective panicles, the total number of spikelets and seed setting rate increased by 4.66% and 2.08%, respectively, thus increasing the rice yield by 5.81% ($p < 0.05$).

Reasonable planting density increased the tillering number and enhanced the lodging resistance of rice [48,49], thereby optimizing rice population light distribution, increasing photosynthetic carbon and N accumulation, promoting the transport of assimilated substances to grains, and improving rice yield and quality [50]. The effective panicle number and kernel number per panicle were significantly increased by CRFDP, and the increase in effective panicle number and kernel number per panicle could increase the capacity of rice pool, thus increasing the grain yield of rice. This study was similar to the study by Yan et al. [51], which showed that the effective panicle number, dry matter accumulation, and yield of rice with the increase in planting density increased by 8.08%, 3.86% and 3.72%, respectively ($p < 0.05$). Zhang et al. [34] showed that the compensation effect of dense planting on rice yield was 83.3% under the 50% reduction of N application. Under the same N application rate, the yield, dry matter weight, and N use efficiency under densification treatment were significantly increased by 6.4%–22.2%, 9.1% and 21.9%, respectively ($p < 0.05$), compared with those under the conventional density. This study was basically consistent with previous studies, and emphasized that densification combined with controlled-release urea could further increase yield and N use efficiency by 26.1%–33.3% and 90.2%–160.0%, respectively($p < 0.05$).

## 4.3. Effects of CRFDP on ammonia volatilization loss in paddy fields and economic sustainability

The experimental results verified the hypothesis that dense planting combined with controlled-release fertilizer (CRF) could reduce ammonia volatilization due to two main reasons. First, after N fertilizer enters the paddy system, its morphology and transformation are affected by various N-transforming microorganisms, mainly consisting of inorganic $NH_4^+$-N and $NO_3^-$-N [52]. In this study, CRBF and CRFDP treatment significantly reduced the concentrations of ammonium N and nitrate N in surface water (Fig 4), decreased the ammonia volatilization rate and ammonia volatilization loss (Fig 3), and decreased the ammonia volatilization intensity during the rice growth period compared with FFP (Tables 6 and 7) ($p < 0.05$). Ammonia volatilization is the main route of N loss in paddy fields, accounting for 70% of total N loss [53]. CRF can slow down N release, coordinate N supply and N demand of rice, achieve rice yield increase and efficient N utilization, and reduce ammonia volatilization [53]. According to Wang et al. [54], the concentration of ammonium N in surface water was significantly positively correlated with the rate of ammonia volatilization, and the concentration of ammonium N in surface water was a key factor to control the loss of ammonia volatilization. Hou et al. [55] found that the CRBF kept the rice yield unchanged, the average ammonium N concentration in surface water decreased by 26.26%–45.61%, and the cumulative ammonia volatilized emissions of early and late rice decreased by 5.2%–38.2% ($p < 0.05$). Li et al. [56] found that ammonia volatilization under slow-release N fertilizer treatment was significantly reduced by 28.7%–30.9% compared with the urea treatment ($p < 0.05$). Second, densification can reduce

ammonia volatilization. The densification treatments are more effective at achieving high N uptake capacity in crops [57], and can enhance the N storage capacity of plant organs [27], achieve high N use efficiency (>62%) and reduce N losses through ammonia volatilization by 9%–17% [58].

Controlled-release fertilizers improve economic benefits. In this study, CRBF increased economic benefits by 41.93% and 28.51% and CRFDP further increased the benefits by 46.87% and 38.25% in 2021 and 2022, respectively, compared with FFP (Table 8) ($p < 0.05$). This was consistent with the findings of a previous study that the application of controlled-release fertilizers improved economic efficiency. Huang et al. [59] stated that the economic benefits of early and late rice increased by 3.00% and 17.03% under the conditions of N reduction and one-time fertilization ($p < 0.05$). Yang et al. [60] showed that, although the cost of controlled-release fertilizer was increased by 30.28%, the profit was increased by more than 9.79% due to the higher yield of rice treated with controlled-release fertilizers ($p < 0.05$). Liu et al. [19] showed that increasing transplanting density and reducing N input could stabilize the rice yield. Zhang et al. [61] found that compared with conventional urea fertilization, the combined application of controlled-release mixed fertilizers increased the income by 2497.8 Yuan ha$^{-1}$, which was an increase of 14.87% ($p < 0.05$). Therefore, the combination of CRBF and densification had significant comprehensive benefits in reducing the fertilizer amount and increasing farmers' income in paddy fields, and thus is a good alternative technology for controlling fertilizer consumption.

## 5. Conclusions

Compared with FFP, CRFDP significantly improved rice yield by 26.1%–33.3%, economic efficiency by 38.25% ~ 46.87%, and N use efficiency by 90.2%–160.0%, while significantly reduced ammonia volatilization intensity by 60.8%–62.5% and ammonia volatilization loss by 46.3%–50.3%. This technology that the combination of controlled-release blended fertilizer and densification may contribute significantly to increasing rice yields and reducing N emissions, offering a viable approach for the green and sustainable development of agriculture with broad application prospects. Further study should be done for multi-point experimental validation to assess the performance under different regions and soil conditions, and to further explore the optimal nitrogen application rate for CRFDP.

## Supporting information

**S1 Table. The cost of agricultural materials, labor, and seedlings.**
(DOCX)

## Acknowledgments

This work is supported by Beijing Academy of Agriculture and Forestry Sciences Innovation Capacity Building Special (KJCX20230304), Key R&D plan of Hubei Province (2023BBB082; 2022BBA002), and Platform Construction of Beijing Academy of Agriculture and Forestry Sciences (PT2023-47). We particularly appreciate the guidance of the editor and reviewers on the refinement of the paper.

## Author contributions

**Conceptualization:** Zijuan Ding, Ren Hu.

**Data curation:** Xiaowei Ma.

**Investigation:** Xuexia Wang, Jun Hou, Guoyuan Zou.

**Methodology:** Jun Hou, Bing Cao.

**Writing – original draft:** Xiaowei Ma.

**Writing – review & editing:** Xiaowei Ma, Jun Hou, Bing Cao.

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
