## [Decision Letter · Decision Letter 0]

19 Nov 2024

PONE-D-24-46000Increasing rice yield with low ammonia volatilization by combined application of controlled-release blended fertilizer and densificationPLOS ONE

Dear Dr. Hou,

Thank you for submitting your manuscript to PLOS ONE. After careful consideration, we feel that it has merit but does not fully meet PLOS ONE’s publication criteria as it currently stands. Therefore, we invite you to submit a revised version of the manuscript that addresses the points raised during the review process.

Reviewer 2 indicated that the statistical methods used are currently vague. A more detailed explanation of the analysis techniques is needed. Specifically, please clarify the types of statistical tests used, the rationale behind selecting them, and how the results are interpreted. Reviewer 3 indicated that the the scientific issue presented in the manuscript is not clearly defined. To improve this, I recommend that the authors clearly outline the research gap in the literature and formulate a well-defined scientific question. This should be done while making sure that the manuscript's language is not overly verbose and lacking conciseness.

We look forward to receiving your revised manuscript.

Kind regards,

Eyas Mahmoud

Academic Editor

PLOS ONE

Journal Requirements:

Reviewers' comments:

Reviewer's Responses to Questions

**Comments to the Author**

1. Is the manuscript technically sound, and do the data support the conclusions?

Reviewer #1: Yes

Reviewer #2: Yes

Reviewer #3: Yes

2. Has the statistical analysis been performed appropriately and rigorously?

Reviewer #1: Yes

Reviewer #2: Yes

Reviewer #3: Yes

3. Have the authors made all data underlying the findings in their manuscript fully available?

Reviewer #1: Yes

Reviewer #2: Yes

Reviewer #3: Yes

4. Is the manuscript presented in an intelligible fashion and written in standard English?

Reviewer #1: Yes

Reviewer #2: Yes

Reviewer #3: No

5. Review Comments to the Author

Reviewer #1: In this study, we combined controlled release fertilizer and dense planting techniques to achieve the dual goal of increasing rice yield and reducing ammonia volatilization.

By comparing the effects of different fertilization and planting density on chlorophyll content, nitrogen transport enzyme activity and dry matter transport in rice, new ways to improve the utilization efficiency of nitrogen fertilizer and reduce the environmental impact are explored, which is somewhat innovative.

Logic is relatively clear, and it is recommended to receive it.

Reviewer #2: Dear Authors,

1. The manuscript presents valuable data on the effects of nitrogen (N) fertilization on rice growth stages; however, the clarity of the figures and tables could be improved. For instance, Figure 4 lacks clear labeling and explanation of the correlation analysis, making it difficult to interpret the relationship between NH4+-N content and ammonia volatilization.

2. The correlation analysis presented in Figure 5 is crucial, yet it lacks a thorough explanation in the text. The authors should provide more context on how the NH4+-N content relates to ammonia volatilization rates, as this is a key aspect of the study

3. There is a noticeable inconsistency in the reporting of statistical significance across different sections. While some results indicate significant differences (p < 0.05) among treatments, others do not provide sufficient context or clarity on the statistical methods used, particularly in the discussion of chlorophyll content and N application rates.

4. The manuscript would benefit from a more comprehensive discussion on the implications of the findings. For example, while the results indicate that N application significantly increased the number of effective panicles and filled grains per panicle, the authors should elaborate on how these findings relate to overall rice yield and agricultural practices.

5. The statistical methods used to analyze the data should be more explicitly stated. For example, the analysis of variance and least significant range mentioned in the results section could benefit from a clearer description of the methodology employed

6. Lastly, the manuscript should ensure that all figures and tables are referenced appropriately in the text. For example, the results presented in Table 2 regarding SPAD values should be discussed in relation to the overall findings of the study to enhance coherence and flow.

Overall, addressing these suggestions will significantly enhance the quality and impact of the manuscript.

Reviewer #3: The scientific issue presented in the text is not clearly defined. Based on literature review, it is necessary to identify the gaps in current research, propose a clear scientific question, and develop a hypothesis to address this question, followed by corresponding solutions.

The discussion should be rewritten as follows: The experimental results validate the hypothesis that high-density planting combined with controlled-release fertilizer (CRF) not only increases yield but also reduces ammonia volatilization. Then, based on the experimental results, delve into explaining how this combination achieves the stated hypothesis.

The language is overly verbose and lacks conciseness, which requires significant improvement.

6. PLOS authors have the option to publish the peer review history of their article (what does this mean? ). If published, this will include your full peer review and any attached files.

**Do you want your identity to be public for this peer review?** For information about this choice, including consent withdrawal, please see our Privacy Policy .

Reviewer #1: No

Reviewer #2: No

Reviewer #3: No

---

## [Author Response · Author response to Decision Letter 1]

14 Dec 2024

Manuscript ID: PONE-D-24-46000

Dear Editors:

We have made revisions to our manuscript "Increasing rice yield with low ammonia volatilization by combined application of controlled-release blended fertilizer and densification (Manuscript ID: PONE-D-24-46000)". Many thanks to the editors and reviewers for their comments and suggestions on the manuscript. In response to these comments and suggestions, we have carefully replied and explained them, and highlighted them in the manuscript.

We submit revised manuscript with changes marked by using the blue colored text. If you have any questions about this manuscript, please don’t hesitate to let me know.

Thank you for your consideration of our manuscript and we look forward to receiving comments from the reviewers. If you have any questions, please do not hesitate to contact me at the following address.

Yours sincerely,

Xiaowei Ma (First author), E-mail: 2022710745@yangtzeu.edu.cn

Jun Hou (Corresponding author), E-mail: houjungoodluck1@163.com

Response to Reviewer #1

Question: In this study, we combined controlled release fertilizer and dense planting techniques to achieve the dual goal of increasing rice yield and reducing ammonia volatilization.

By comparing the effects of different fertilization and planting density on chlorophyll content, nitrogen transport enzyme activity and dry matter transport in rice, new ways to improve the utilization efficiency of nitrogen fertilizer and reduce the environmental impact are explored, which is somewhat innovative.

Logic is relatively clear, and it is recommended to receive it.

Reply: Thanks to the reviewer for his recognition of this paper, we have further revised and improved the manuscript, supplemented the determination methods of SPAD values and N transferase activity, and conducted a more in-depth discussion on the promotion of rice yield and emission reduction by controlled-release fertilizer combined with density and nitrogen reduction, hoping to be more in line with the requirements of the journal.

Response to Reviewer #2

Question 1: The manuscript presents valuable data on the effects of nitrogen (N) fertilization on rice growth stages; however, the clarity of the figures and tables could be improved. For instance, Figure 4 lacks clear labeling and explanation of the correlation analysis, making it difficult to interpret the relationship between NH4+-N content and ammonia volatilization.

Reply: We noticed this problem, so in the revised manuscript, we added the markup to Figure 4 and added the explanation of the significance analysis below Figure 4.

Question 2: The correlation analysis presented in Figure 5 is crucial, yet it lacks a thorough explanation in the text. The authors should provide more context on how the NH4+-N content relates to ammonia volatilization rates, as this is a key aspect of the study.

Reply: We apologize for the lack of background on the relationship between NH4+-N content and ammonia volatility in this article, so we have supplemented and improved this in introduction, i.e., “Nitrogen fertilizer applied to farmland will significantly increase the surface water NH4+-N concentration (Liu et al., 2015), and it will be lost through NH3 volatilization ways (Hou et al., 2018).” Please see Line66-68.

And at the same time discussed in depth in the discussion, i.e., “The experimental results verified the hypothesis that dense planting combined with controlled-release fertilizer (CRF) could reduce ammonia volatilization due to two main reasons. First, after N fertilizer enters the paddy system, its morphology and transformation are affected by various N-transforming microorganisms, mainly consisting of inorganic NH4+-N and NO3-N (Xiao et al., 2023).” Please see Line 489-492

Question 3: There is a noticeable inconsistency in the reporting of statistical significance across different sections. While some results indicate significant differences (p < 0.05) among treatments, others do not provide sufficient context or clarity on the statistical methods used, particularly in the discussion of chlorophyll content and N application rates.

Reply: We added more detailed methods for SPAD and nitrogen transferase sampling and determination, i.e., “At the tillering, booting, and heading stages of rice, 10 functional leaves were randomly selected from each plot, with 5 points taken from each leaf, and the relative chlorophyll content SPAD value was measured using the chlorophyll analyzer SPAD-502 (Konica Minolta, Japan). At these stages, 10 functional leaves were taken from each plot, and the activities of nitrate reductase (NR) and glutamine synthetase (GS) were determined using the in vitro method. The activities of glutamate dehydrogenase (GDH) and glutamate synthetase (GOGAT) were measured using biochemical kits (Suzhou Keming Biotechnology Co., Ltd.).” Please see Line150-156. Significant differences (p < 0.05) between treatments were noted in the discussion.

Question 4: The manuscript would benefit from a more comprehensive discussion on the implications of the findings. For example, while the results indicate that N application significantly increased the number of effective panicles and filled grains per panicle, the authors should elaborate on how these findings relate to overall rice yield and agricultural practices.

Reply: We supplemented the controlled-release fertilizer to ensure N supply and promote the increase of the effective number of effective panicle number and filled grains per panicle, i.e., “CRFDP treatment significantly increased NRE by 18.1%–21.7% compared with CRBF(p<0.05). The main reason was that CRFDP treatment significantly increased the effective panicles by 26.1% and 27.2%, and the grains per panicle by 18.7% and 13.4% (2021 and 2022) (p<0.05) compared with FFP treatment. This is because the N supply of CRBF was superior to that of soluble N fertilizers (Huang et al., 2019). The replacement of urea by controlled-release urea could significantly increase the effective panicles and the grains per panicle,” Please see Line457-462.

Question 5: The statistical methods used to analyze the data should be more explicitly stated. For example, the analysis of variance and least significant range mentioned in the results section could benefit from a clearer description of the methodology employed

Reply: We refined more explicit statistical methods for analyzing data in Materials and Methods, i.e., “SPSS 19.0 (IBM, Inc., Armonk, NY, USA) was used for a one-way analysis of variance (ANOVA), and significant differences (p < 0.05) between the treatments are indicated by different letters. All the data diagrams were prepared using Microsoft Excel 2010 (Redmond, WA, USA), and the correlation between NH3 volatilization and the concentration of NH4+-N in field water and the correlation between NH3 volatilization and concentration of NH4+-N in soil were analyzed. Diagrams for the diffusion of NH4+-N in the soil were prepared using Surfer 8.0 (Surface mapping system; Golden Software, Inc., Golden, CO, USA).” Please see Line225-231.

Question 6: Lastly, the manuscript should ensure that all figures and tables are referenced appropriately in the text. For example, the results presented in Table 2 regarding SPAD values should be discussed in relation to the overall findings of the study to enhance coherence and flow.

Reply: We revised the manuscript, added tables and figures of references in the discussion section, and combined the results of the SPAD value with the overall results of the study in the discussion, i.e., “CRF continuously supplies nitrogen to meet the nitrogen demand during the growth period of rice (Zhang, et al., 2021). At the same time, densification increases the rice root system and can more fully absorb inorganic N, increase the number of rice tillers 18.16-19.24%, increase rice SPAD 2.54-3.38%, ultimately increase rice yield 6.01-7.71%(p<0.05) (Dong, et al., 2023; Sun, et al., 2024).” Line406-409.

We will discuss more results with each other, i.e., “CRFDP treatment significantly increased NUE by 18.1%–21.7% compared with CRBF(p<0.05). The main reason was that CRFDP treatment significantly increased the effective panicles by 26.1% and 27.2%, and the grains per panicle by 18.7% and 13.4% (2021 and 2022) (p<0.05) compared with FFP treatment. This is because the N supply of CRBF was superior to that of soluble N fertilizers (Huang et al., 2019). The replacement of urea by controlled-release urea could significantly increase the effective panicles and the grains per panicle,” Please see Line457-462.

Response to Reviewer #3

Question 1: The scientific issue presented in the text is not clearly defined. Based on literature review, it is necessary to identify the gaps in current research, propose a clear scientific question, and develop a hypothesis to address this question, followed by corresponding solutions.

Reply: Thank you for your comments. Based on the characteristics of controlled-release fertilizer and its role in increasing yield and reducing emissions through densification, we proposed the hypothesis in the introduction, that the combination of controlled-release fertilizer with densification can promote increased yield and reduced emissions, and put forward a feasible research plan. i.e., “Based on the role of controlled-release fertilizer and the yield increase caused by densification with reduced nitrogen, we speculated that combined application of controlled-release blended fertilizer and densification could increase rice yield and reduce ammonia volatile emissions, and improve economic benefits after reducing the N input.” Please see Line100-103.

The explanation is as follows:

“The main reason was that CRFDP significantly increased the effective panicles by 26.1% and 27.2%, and the grains per panicle by 18.7% and 13.4% (2021 and 2022) (p<0.05) compared with FFP. This is because the N supply of CRBF was superior to that of soluble N fertilizers (Huang et al., 2019).” Please see Line458-461.

And, “The experimental results verified the hypothesis that dense planting combined with controlled-release fertilizer (CRF) could reduce ammonia volatilization due to two main reasons. First, after N fertilizer enters the paddy system, its morphology and transformation are affected by various N-transforming microorganisms, mainly consisting of inorganic NH4+-N and NO3-N (Xiao et al., 2023)” Please see Line489-492.

And, “Second, densification can reduce ammonia volatilization. The densification treatments are more effective at achieving high N uptake capacity in crops(Yu et al., 2023), and can enhance the N storage capacity of plant organs (Sun, et al., 2024), achieve high N use efficiency (>62%) and reduce N losses through ammonia volatilization by 9%-17% (Dawar et al., 2024).” Please see Line506-509.

Question 2: The discussion should be rewritten as follows: The experimental results validate the hypothesis that high-density planting combined with controlled-release fertilizer (CRF) not only increases yield but also reduces ammonia volatilization. Then, based on the experimental results, delve into explaining how this combination achieves the stated hypothesis.

Reply: The discussion was modified and the results validated the hypothesis that controlled-release fertilizer combined with densification would increase yield and reduce ammonia volatilization. The assumptions are explained based on the results of the implementation, i.e., “CRF continuously supplies N to meet the N demand during the growth period of rice (Zhang, et al., 2021). At the same time, densification increases the rice root system and can more fully absorb inorganic N, increase the number of rice tillers 18.16-19.24%, increase rice SPAD 2.54-3.38%, ultimately increase rice yield 6.0107.71%(p<0.05) (Dong, et al., 2023; Sun, et al., 2024).” Please see Line406-409.

And, “The experimental results verified the hypothesis that dense planting combined with controlled-release fertilizer (CRF) can increase yield and N uptake.” Please see Line449-450.

And, “The experimental results verified the hypothesis that dense planting combined with controlled-release fertilizer (CRF) could reduce ammonia volatilization due to two main reasons. First, after N fertilizer enters the paddy system, its morphology and transformation are affected by various N-transforming microorganisms, mainly consisting of inorganic NH4+-N and NO3-N (Xiao et al., 2023).” Please see Line489-492.

And, “Second, densification can reduce ammonia volatilization. The densification treatments are more effective at achieving high N uptake capacity in crops(Yu et al., 2023), and can enhance the N storage capacity of plant organs (Sun, et al., 2024), achieve high N use efficiency (>62%) and reduce N losses through ammonia volatilization (9-17%) (Dawar et al., 2024).” Please see Line506-509.

Question 3: The language is overly verbose and lacks conciseness, which requires significant improvement.

Reply: We have made modifications to the following.

1) In the abstract, we have modified the “Compared with FFP treatment, CRFDP treatment significantly increased yield by 33.3% and 26.1% and economic benefit by 46.9% and 38.3% in 2021 and 2022, respectively, at a 16.7% reduction in N dosage.” To “Compared with FFP, CRFDP with 16.7% reduction of N input significantly increased yield by 33.3% and 26.1% and economic benefit by 46.9% and 38.3% in 2021 and 2022, respectively.”

2) In the abstract, we have modified the “Compared with FFP treatment, CRFDP treatment reduced ammonia volatilization intensity by 62.5% and 60.8% and cumulative ammonia volatilization loss by 46.3% and 50.3% by reducing 69.0%–93.8% and 57.8%–89.7% NH4+-N of surface water in 2021 and 2022, respectively.” to “Compared with FFP, CRFDP reduced ammonia volatilization intensity by 62.5% and 60.8%, cumulative ammonia volatilization loss by 46.3% and 50.3% and also lowered NH4+-N of surface water by 69.0%–93.8% and 57.8%–89.7% in 2021 and 2022, respectively.”

3) In the introduction, we have modified the “Therefore, effective measures should be taken to reduce N loss in farmland to maintain crop yield.” to “Therefore, effective measures should be taken to reduce N loss and maintain high crop yield.”

4) In the introduction, we have modified the “Zhu et al. (2019) demonstrated that densification treatment increases planting density by about 50%, correspondingly reduces nitrogen application rate by about 30%, increases NUE by 14.3-50.6%, and increases rice yield by 0.5-7.4%.” to “Zhu et al. (2019) demonstrated that densification increases planting density by about 50%, correspondingly reduces N input by about 30%, and increases rice yield and NUE by 0.5-7.4% and 14.3-50.6%.”

5) In the introduction, we have modified the “Appropriately increased densification is a good cultivation technology model to achieve high yield, use N fertilizer efficiently, and reduce N emission of rice (Zhong et al., 2023).” to “Appropriately increased densification is a good cultivation technology model to achieve high rice yield and high N use efficiency and reduce N loss (Zhong et al., 2023).”

6) In the results 3.1, we have modified the “In the tillering stage, the GDH activity was higher after the FFP treatment than after the other three N application treatments. In the booting and full heading stages, the GDH activity was significantly higher after CRBF treatment than after the other three N application treatments.” to “At the tillering stage, FFP had the higher GDH activity than the other three N application treatments. At the booting and full heading stages, CRBF had significantly higher GDH activity than the other three N application treatments.”

7) In the results 3.1, we have modified the “In the tillering and booting stages, the NR activity was higher in rice leaves treated with FFP than in those treated with the other three optimized N applications. In the full heading stage, the NR activity was significantly higher after the CRBF and CRFDP treatments than after the FFP and OPT treatments.” to “At the tillering and booting stages, FFP had higher NR activity than the other three optimized N applications. At the full heading stage, the CRBF and CRFDP had higher NR activity than FFP and OPT.”

8) In the results 3

---

## [Decision Letter · Decision Letter 1]

13 Jan 2025

Increasing rice yield with low ammonia volatilization by combined application of controlled-release blended fertilizer and densification

PONE-D-24-46000R1

Dear Dr. Hou,

We’re pleased to inform you that your manuscript has been judged scientifically suitable for publication and will be formally accepted for publication once it meets all outstanding technical requirements.

Kind regards,

Eyas Mahmoud

Academic Editor

PLOS ONE

Additional Editor Comments (optional):

Reviewers' comments:

Reviewer's Responses to Questions

**Comments to the Author**

1. If the authors have adequately addressed your comments raised in a previous round of review and you feel that this manuscript is now acceptable for publication, you may indicate that here to bypass the “Comments to the Author” section, enter your conflict of interest statement in the “Confidential to Editor” section, and submit your "Accept" recommendation.

Reviewer #2: All comments have been addressed

Reviewer #3: All comments have been addressed

2. Is the manuscript technically sound, and do the data support the conclusions?

Reviewer #2: Yes

Reviewer #3: Yes

3. Has the statistical analysis been performed appropriately and rigorously?

Reviewer #2: Yes

Reviewer #3: Yes

4. Have the authors made all data underlying the findings in their manuscript fully available?

Reviewer #2: Yes

Reviewer #3: Yes

5. Is the manuscript presented in an intelligible fashion and written in standard English?

Reviewer #2: Yes

Reviewer #3: Yes

6. Review Comments to the Author

Reviewer #2: Dear Authors,

Thank you for following the suggestions. The manuscript now meets the required standards. Well done.

Reviewer #3: The issues raised by this reviewer have been adequately addressed, and the manuscript is ready for publication.

7. PLOS authors have the option to publish the peer review history of their article (what does this mean? ). If published, this will include your full peer review and any attached files.

**Do you want your identity to be public for this peer review?** For information about this choice, including consent withdrawal, please see our Privacy Policy .

Reviewer #2: **Yes: ** Dr. Niaz Ali

Reviewer #3: No

---

## [Editor Report · Acceptance letter]

PONE-D-24-46000R1

PLOS ONE

Dear Dr. Hou,

I'm pleased to inform you that your manuscript has been deemed suitable for publication in PLOS ONE. Congratulations! Your manuscript is now being handed over to our production team.

Kind regards,

on behalf of

Dr. Eyas Mahmoud

Academic Editor

PLOS ONE